# COVID-19 Vaccine: A Potential Risk Factor for Accelerating the Onset of Bullous Pemphigoid

**DOI:** 10.3390/vaccines12091016

**Published:** 2024-09-05

**Authors:** Anna Pira, Feliciana Mariotti, Francesco Moro, Biagio Didona, Giovanni Luca Scaglione, Annarita Panebianco, Damiano Abeni, Giovanni Di Zenzo

**Affiliations:** 1Molecular and Cell Biology Laboratory, Istituto Dermopatico dell’Immacolata (IDI)-IRCCS, 00167 Rome, Italy; f.mariotti@idi.it (F.M.); f.moro@idi.it (F.M.); g.dizenzo@idi.it (G.D.Z.); 2Dermatology Unit, Istituto Dermopatico dell’Immacolata (IDI)-IRCCS, 00167 Rome, Italy; 3Rare Diseases Unit, Istituto Dermopatico dell’Immacolata (IDI)-IRCCS, 00167 Rome, Italy; b.didona@idi.it; 4Bioinformatic Unit, Istituto Dermopatico dell’Immacolata (IDI)-IRCCS, 00167 Rome, Italy; g.scaglione@idi.it; 5Medical Direction, Istituto Dermopatico dell’Immacolata (IDI)-IRCCS, 00167 Rome, Italy; a.panebianco@idi.it; 6Clinical Epidemiology Unit, Istituto Dermopatico dell’Immacolata (IDI)-IRCCS, 00167 Rome, Italy; d.abeni@idi.it

**Keywords:** autoimmune bullous disease, bullous pemphigoid, COVID-19, pandemic, SARS-CoV-2, vaccine

## Abstract

Bullous pemphigoid (BP) is the most common autoimmune bullous disease, whose main autoantigens are hemidesmosomal components BP180 and BP230. Although recent studies found no association between COVID-19 vaccines and BP, since mass vaccinations started, more than 90 vaccine-associated BP cases have been reported. To find an agreement among real-life clinical observations and recent epidemiologic data, we further investigated this topic. A total of 64 patients with BP onset in 2021 were demographically, clinically, and serologically characterized: 14 (21.9%) vaccine-associated patients (VA) developed BP within 5 weeks from the first/second vaccine dose. VA and vaccine-non-associated (VNA) patients had similar demographics and clinical and immunological characteristics. Noteworthy, the monthly distribution of BP onset during mass vaccinations paralleled vaccine administration to the elderly in the same catchment area. Additionally, in 2021, BP onsets in April–May and June–July significantly increased (*p* = 0.004) and declined (*p* = 0.027), respectively, compared to the three years before vaccination campaigns (2018–2020). Interestingly, VA and VNA patients showed statistically significant differences in the use of inhalers and diuretics. Our findings suggest that the COVID-19 vaccine may constitute an accelerating factor that, together with other triggering factors, could act in genetically predisposed individuals with possible sub-clinical autoreactivity against BP antigens, slightly accelerating BP onset.

## 1. Introduction

Bullous pemphigoid (BP) is the most common autoimmune blistering disease (AIBD) [1]. It characteristically affects the elderly, with intense pruritic eruptions accompanied by the formation of subepidermal blisters due to the presence of tissue-bound and circulating autoantibodies (autoAbs) targeting distinct components of the skin [2]. Specifically, BP autoAbs are directed against BP180 and BP230, which are components of hemidesmosomes, junctional adhesion complexes that promote dermal-epidermal cohesion [1]. BP has a complex and largely unknown etiopathogenesis, and it has been acknowledged that multiple trigger factors may induce events that lead to subepithelial blister formation [3]. Among specific trigger factors, drugs and vaccines were reported to be associated with the onset of several BP cases [3]: in 2020, a systematic review listed the vaccines for swine flu, herpes zoster virus, influenza, tetanus toxoid and tetracoq, rotavirus, and the varying constituents of the hexavalent combined vaccines as possibly implicated in the development of drug-induced BP [4].

The coronavirus disease 2019 (COVID-19) pandemic had a severe public health impact in many countries. In Italy, the first two cases of COVID-19 were identified on 30 January 2020. The progressive increase in cases led to a generalized lockdown of approximately 2 months. The mass vaccination campaign began across Europe on 27 December 2020, and still continues today [5,6]. The pandemic, in addition to having produced direct damage due to the SARS-CoV-2 infection, has produced indirect damage because of the impact on the care of patients suffering from other conditions, who have seen delays in their diagnosis and treatment [6,7]. In addition, although the mass vaccination campaign has largely mitigated the pandemic’s damage, its effect on the immune system could have had a role in the exacerbation or induction of several autoimmune diseases [8,9,10]. Numerous cases of autoimmune disorders arising after COVID-19 vaccine administration, especially mRNA vaccines, have been reported in the literature, including, among others, systemic lupus erythematosus, psoriasis, rheumatoid arthritis, and AIBDs [11,12,13,14,15,16,17,18,19,20,21,22,23,24,25] (Appendix A). Several studies have investigated the effects of these vaccines on the immune system and the possible mechanisms leading to autoimmunity. Enhanced TGF-beta signaling and increased NF-kB responses, as well as an increased expression of IL-6 and TNF-alfa, have been observed following COVID-19 vaccination [26,27]. The release of pro-inflammatory cytokines (such as IL-6) after COVID-19 vaccination may dysregulate the T cell response, which is largely involved in the development of BP [28,29]. Moreover, immune-senescence could be associated both with COVID-19 and with COVID-19 vaccination [30,31], suggesting a possible role of this phenomenon in causing the induction of autoimmune diseases, especially BP [3,32]. A recent review reports that since the beginning of mass vaccinations against COVID-19, more than 92 BP cases of new-onset and many cases of relapse or worsening of pre-existing BP have been published [33]. However, despite the long-lived suspicion about a possible role of vaccines on autoimmunity development, to this day, the causal association between vaccinations and the development of chronic autoimmune diseases has not been fully demonstrated [34]. As for COVID-19 vaccines, a recent study challenges the causal relationship between the new onset of an AIBD and COVID-19 vaccination, reporting that there was no cross-reactivity between circulating anti-SARS-CoV-2 antibodies and AIBD autoantigens [35]. A recent study by Birabaharan et al. compared very large cohorts of individuals who did or did not receive a mRNA COVID-19 vaccine and found no increased risk of developing BP in vaccinated people, suggesting that there is no causal association between BP onset and COVID-19 vaccination [36]. Nonetheless, the many valuable real-world clinical data pointing to the necessity to further investigate this issue should be taken into consideration. Thus, our study aims to provide additional information on the relationship between SARS-CoV-2 vaccines and BP onset, investigating whether the vaccine may be considered a risk factor that may shorten the time to BP onset in predisposed individuals.

## 2. Materials and Methods

### 2.1. Study Population

Sixty-four incident cases of BP were enrolled in our clinic in 2021 and 2022, according to the following eligibility criteria: (i) age of 18 years or older; (ii) clinical diagnosis consistent with BP; (iii) histological exam compatible with BP; (iv) positive direct immunofluorescence (DIF) and, in case of negative or unavailable result, positive indirect immunofluorescence (IIF) and/or ELISAs detecting IgG for BP180/BP230 (with a cut-off value of ≥9 U/mL) was considered; (v) BP onset in 2021. Clinical data were collected face-to-face during baseline or follow-up visits in our Dermatology Units and through electronic charts. All data regarding COVID-19 vaccine administration in the Lazio Region, Italy (i.e., which includes the city of Rome) were obtained by the Commissioner’s organizational structure for COVID-19 emergency on GitHub public repository [37]. We considered a latency time of 5 weeks as sufficient time both for stimulation of pre-existent autoimmunity and for the induction of a novel autoimmune response triggered by the vaccine. Moreover, as previously shown, a one-month latency period from the time of vaccination may be appropriate for an autoimmune response to hemidesmosomal antigens to be developed [38]. Therefore, patients were considered vaccine-associated (VA) when the latency to lesions was up to 5 weeks after the administration of the first or second dose of COVID-19 vaccine, while they were considered vaccine-non-associated (VNA) if the lesions appeared before vaccination, more than 5 weeks after the first or second COVID-19 vaccine dose, and, obviously, if the patients were not vaccinated.

The local ethics committee approved this study, which was conducted in accordance with the Declaration of Helsinki guidelines. All patients gave written informed consent.

### 2.2. Enzyme-Linked Immunosorbent Assays

Anti-BP180 and anti-BP230 IgG autoAbs were detected using commercial ELISA kits (MBL International, Woburn, MA, USA) following the manufacturer’s instructions, with a cut-off value of ≥9 U/mL. Commercial ELISA kits only detect autoAbs targeting the immunodominant region of BP180 (NC16A); however, autoantibody reactivity toward BP180 is not restricted to NC16A. To better characterize their immunological profile, patient sera were also tested to detect IgG autoAbs directed against other epitopes located in the mid-portion (E-1080) and C-terminus (E-1331) regions of BP180 ectodomain by using an ELISA with GST-1080/1331, as previously described [39].

### 2.3. Direct and Indirect Immunofluorescence

DIF results were obtained on sections stained with fluorescein isothiocyanate-conjugated goat anti-human Ig (IgG, IgA, and IgM) and C3 (Kallestad Diagnostic, Chaska, MN, USA) under a fluorescence microscope. Slides containing human intact or salt-split skin were used to perform IIF as described [40].

### 2.4. Statistical Analyses

Data regarding VA and VNA patients and BP onset were analyzed using the Fisher’s exact (probability) test, the Mann-Whitney U test, and Spearman correlation coefficient using GraphPad Prism software version 9.4.1 for Windows. The binary logistic regression was performed using the latest version of R (4.3.3), utilizing the base R packages. The binary logistic regression models, including data preprocessing, model fitting, and results interpretation, were conducted within the RStudio environment. Data reduction analysis, Principal Component Analysis (PCA), and Heatmap plot were all performed using the latest docker version of ClustVis software [41]. Differences were considered significant when the *p* value was ≤0.050.

## 3. Results

### 3.1. In 22% of Patients, Bullous Pemphigoid Onset Was Temporally Associated with COVID-19 Vaccination

Of the 64 study participants, 14 (21.9%) were considered VA, and 50 (78.1%) were considered VNA. Among VNA patients, 16 (32.0%) developed the lesions before vaccination, 29 (58.0%) more than 5 weeks after the first or second COVID-19 vaccine dose, and 5 (10.0%) were not vaccinated. Our study population was characterized by a slight male predominance, both in VA and VNA patients (M/F ratio: 1.3 in both groups). The two groups did not differ significantly in terms of mean age (76.7 and 80.0 years in VA and VNA patients, respectively; *p* = 0.178) (Table 1). As for the different vaccines, in regards to the first two doses, of 14 VA patients, 10 (71.4%) received the COMIRNATY Pfizer-BioNTech vaccine, 3 (21.4%) received the Moderna mRNA-1273 vaccine, and 1 (7.1%) developed BP after the first dose of the ChAdOx1/nCoV-19-AstraZeneca/Vaxzevria vaccine and received the second dose with the Pfizer vaccine. Of the 45 vaccinated VNA patients, in 42, the type of vaccine administered was known. Thirty-five VNA patients (83.3%) received the COMIRNATY Pfizer-BioNTech vaccine, 5 (11.9%) received the Moderna mRNA-1273 vaccine, and 2 (4.8%) the ChAdOx1/nCoV-19-AstraZeneca/Vaxzevria vaccine. The differences in the distribution of the vaccines in VA and VNA patients and mRNA vaccine compared to virus-based (Astra-Zeneca) were not significant (Pfizer: *p* = 0.439; Moderna: *p* = 0.398; AstraZeneca: *p* > 0.999; mRNA vs. AstraZeneca: *p* > 0.999) (Table 1). Among VA patients, eight (57.1%) developed BP after the first dose, with a mean latency to BP onset of 12.0 (SD ± 4.4) days and with a median latency to BP onset of 11.3 (IQR: 7.0–16.1) days; and six (42.9%) developed BP after the second dose, with a mean latency to BP onset of 14.4 (SD ± 3.9) days and with a median latency to BP onset of 15.0 (IQR: 12.6–17.0) days after the second dose (Table 1).

### 3.2. Immunological Profile of Vaccine-Associated and Non-Associated Patients Was Similar

A total of 42 of 64 BP patients (8 VA and 34 VNA, of whom 5 were not vaccinated) were enrolled at diagnosis and had not yet been treated at the time when the blood sample was collected, so their immunological profiles were investigated within the framework of our study. VA patients showed no significant differences in terms of reactivity to all antigens/epitopes analyzed when compared to the VNA group. Specifically, in regard to BP180, 7 VA patients (87.5%) and 27 VNA patients (79.4%) were positive, with a median titer of 259.9 U/mL and 88.1 U/mL, respectively (Table 2). As for BP230, 5 VA patients (62.5%) and 16 VNA patients (47.0%) were positive, with a median titer of 60.5 U/mL and 46.5 U/mL, respectively (Table 2). Furthermore, no major differences between VA and VNA patients were found in terms of reactivity to both E-1080 (37.5% vs. 44.1%) and E-1331 (25.0% vs. 41.1%). (Table 2). The only significant differences when considering autoAbs titers were observed for E-1080 (median titer: 15.7 PIV in VA and 95.8 PIV in VNA, *p* = 0.039) and E-1331 (median titer: 138.8 PIV in VA and 46.9 PIV in VNA, *p* = 0.033).

To assess the role of COVID-19 vaccinations in any possible differences in immunological profiles, the five VNA non-vaccinated patients were compared not only to VA patients but also to all the vaccinated patients, either classified as VA or VNA, enrolled at diagnosis. The comparison between autoAbs reactivity and titers showed no statistically significant differences (Appendix A).

Beyond immunological profiles, disease activity can also be measured with the Bullous Pemphigoid Disease Area Index (BPDAI) score, which was proposed in 2012 by an international consensus of experts with the aim of providing a specific tool to objectively assess disease severity [42]. BPDAI scores of VA and VNA patients were comparable (mean: 41.1 and 37.2, respectively; *p* = 0.669) (Table 2). Also, no differences were found in the comparison between VA, vaccinated, and non-vaccinated patients (Appendix A).

For 27 patients enrolled at disease diagnosis who developed BP after COVID-19 vaccination (both VA and VNA), Spearman’s correlation coefficient was computed to assess the relationship between the time of latency from the first dose of vaccine to BP onset and autoAbs titers. Interestingly, BP180 (rs(25) = −0.37, 95% CI [−0.66, 0.03], *p* = 0.062) and BP230 (rs(25) = −0.40, 95% CI [−0.68, −0.01], *p* = 0.040) autoAbs titers seem to be inversely correlated to the time of latency, while not pathogenic autoAbs to E-1080 and E-1331 showed no correlation (Appendix A).

### 3.3. The Onset of Bullous Pemphigoid Paralleled COVID-19 Mass Vaccinations

A possible link between BP onset and vaccination was further investigated by comparing the monthly distribution of BP onset throughout 2021 in our study population and COVID-19 vaccination in the general population over 70 years old. Interestingly, the onset of BP in 2021 showed a sharp increase in May, following the time when most of the elderly population received the first/second dose of the COVID-19 vaccine, which reached its peak in April (Figure 1). A Spearman’s correlation coefficient was computed to evaluate the relationship between vaccinations and BP onset, showing a positive correlation between the two variables, although it was statistically non-significant: rs(10) = 0.53, 95% CI [−0.08, 0.85] (*p* = 0.078).

### 3.4. Monthly Distribution of Bullous Pemphigoid Onset during Mass Vaccinations Is Peculiar and Differs from the Previous Years

The monthly distribution of BP onset in the 64 study participants was compared to that of 99 patients who visited our Institute in the 3 years preceding mass vaccinations (i.e., 2018–2019–2020) (Figure 2). Spearman’s correlation seems to indicate no linear relationship between the distributions of BP onset in 2021 and in previous years (rs(10) = 0.05, 95% CI [−0.54, 0.61], *p* = 0.873). Interestingly, in 2021, there was a statistically significant increase in BP onset in April and May (30% vs. 11% in 2018–2020, *p* = 0.004) and a significant decrease in June and July (11% vs. 25% in 2018–2020, *p* = 0.027) (Figure 2). Moreover, despite the linear relationship of BP onset in VA e VNA patients (rs(10) = 0.59, 95% CI [0.00, 0.87], *p* = 0.049), the observed increase in BP in April and May was more pronounced among VA patients compared to VNA patients, although statistically non-significant (*p*= 0.096) (Figure 3).

### 3.5. Other Trigger Factors Possibly Involved in Bullous Pemphigoid Onset

The possible role of other trigger and predisposing factors, such as drug intake and comorbidities, in the induction of BP was investigated. Of note, some drugs seemed to facilitate BP onset: VA patients had a higher intake of antiplatelet agent clopidogrel (28.6% vs. 8%, *p* = 0.062) and anticonvulsants (28.6% vs. 8%, *p* = 0.062) and a statistically significant association with the use of inhalers (14.3% vs. 0%, *p* = 0.045) compared to VNA ones (Table 3). Interestingly, a significant difference was also found in the opposite direction: VNA patients had a higher intake of diuretics compared to VA ones (50% vs. 14%, *p* = 0.030) (Table 3). To uncover patterns and relationships within the pharmacological dataset, a data-reduction strategy, including PCA and heatmap, was used. PCA was employed to explore the underlying patterns and relationships among three distinct groups of patients (NV, not vaccinated; VA, vaccine-associated; and VNA-NV, vaccine-non-associated-NV BP patients) based on their pharmacological treatments. The dataset included five different classes of drugs, which were used as variables in the analysis. By reducing the dimensionality of the data, PCA (Appendix A, panel A) allowed for the visualization of the variance explained by each principal component, facilitating the identification of potential clusters or overlapping distributions among the patient groups. Even if this approach explained about 19% of the first component variance, we found a narrow overlap among patients without showing any specific cluster for any of the three groups. Furthermore, to assess potential recurrent associations among the drugs in this cohort, we visualized and analyzed the data using a heatmap (Appendix A, Panel B). Unfortunately, the small size of the study population hinders the production and interpretation of conclusive data on drugs, and we believe studies with larger cohorts are necessary to discuss the impact of drugs on vaccine-associated BP. We were unable to perform further statistical analyses due to the small sample size, and additional data are required to fulfill this task.

No major differences in terms of comorbidities in the two groups were observed (Appendix A). To assess whether the levels of specific autoAbs could predict the presence of various comorbidities in VA and VNA groups, a binary logistic regression was applied. Despite the comprehensive analysis, the results indicated that none of the autoAb titers were statistically significant predictors of the comorbidities in either the VNA or VA groups. This suggests that there is no meaningful association between the autoAbs levels and the presence of the examined comorbidities in the studied population (Appendix A).

## 4. Discussion

Discordant data have been published about the association between SARS-CoV-2 vaccination and BP onset. On the one hand, Birabaharan et al. found no difference in the risk of BP onset among COVID-19-vaccinated individuals and unvaccinated ones [36]. In line with this data, a recent study showed that circulating anti-SARS-CoV-2 antibodies do not cross-react with AIBD autoantigens [35]. However, molecular mimicry and the resulting cross-reactivity are not the only mechanisms that may be able to trigger BP after COVID-19 vaccination: epitope spreading, “bystander activation” of self-reactive lymphocytes, polyclonal activation due to adjuvant reaction, and somatic mutation of immunoglobulin variable genes have all been described as possible mechanisms leading to infection- or vaccine-induced autoimmunity [10]. In addition, as BP onset after COVID-19 vaccination is a very rare event, even in the case of an actual causal link, Birabaharan and coworkers should have considered over 2 billion individuals in each exposure group in order to obtain a significant difference in terms of disease incidence [43]. To date, the link between BP onset and COVID-19 vaccination is supported by many real-world clinical data, and we believe that our results, paired with well-designed future studies aimed at better understanding this association, should be considered before drawing any conclusions.

The present study reports 64 cases of new-onset BP, 14 of whom (VA) (22%) developed the disease temporally associated with COVID-19 vaccination. The limitations are represented by (i) the small sample size, especially of non-vaccinated individuals, and (ii) the absence of follow-up visits, which did not allow the determination of possible differences throughout the disease course in different groups of patients.

There was no significant difference in VA and VNA patients in terms of mean age (76.7 vs. 80.0 years) and M/F ratio. In addition, the mean age at onset in VA patients does not significantly differ from published data on vaccine-associated BP (75.3 years) [33]. Similarly, the M/F ratio (1.3) is very similar to the one observed in previously published cases (i.e., 1.4) [33], suggesting demographics are probably not involved in BP onset following vaccination.

As for latency to BP onset, the patients in our study seem to be similar to those in other published case series. Sixty-nine published cases with BP onset after the first or second vaccine dose reported a mean latency of 12.2 (SD ± 11.9) days from the first dose (35 patients) and 8.3 (SD ± 6.1) days from the second dose (34 patients) [33], whereas in this study, the mean latency from the first and second dose was 12.0 (SD ± 4.4) and 14.4 (SD ± 3.9) days, respectively.

The differences in vaccine type administration in VA and VNA patients were not significant, seemingly indicating that vaccine types are not associated with different risks of developing BP. However, the association with a specific vaccine would need larger cohorts standardized by specific vaccine administration to be accurately assessed. VA patients had a typical clinical phenotype of BP, with intense pruritus and tense bullae on an erythematous base and BPDAI scores similar to VNA patients. Moreover, the BPDAI scores of the five non-vaccinated individuals were comparable to the ones of the VA and all vaccinated patients. Regarding immunological profiles, the frequency of reactivity to BP180 and BP230 was similar between VA and VNA (87.5% vs. 79.4% for BP180 and 62.5% vs. 47.0% for BP230) and in line with reactivity assessed in previous studies (77% for BP180 and 45% for BP230) [33]. In addition, IgG titers for BP180 and BP230 were similar in the two groups. As for the BP180 epitopes (E-1080 and E-1331), the reactivity was also similar in VA and VNA patients, while a significant difference was found in the titers of E-1080 (which was higher in VNA patients) and E-1331 (higher in VA patients). However, as median titers were calculated on positive sera only, it is worth noting that these data were obtained from an extremely small number of VA patients (three and two patients). Also, the comparison of non-vaccinated patients with VA and vaccinated individuals showed no statistically significant differences, further suggesting that vaccination does not impact the immunological profiles of BP patients.

The present study is the first to describe the monthly distribution of BP onset in the year of COVID-19 mass vaccinations (2021) while comparing it with the monthly distribution of vaccine administration in the same catchment area. Noteworthy, the increase in BP onset percentage in May 2021 was immediately following the time when most of the elderly population of the Lazio Region received their first/second dose of the COVID-19 vaccine, further supporting a possible causal link between vaccination and BP onset. Although statistically not significant, a positive correlation between vaccinations and BP onset was found (rs(10) = 0.53, 95% CI [−0.08, 0.85] (*p* = 0.078).

A previous study by D’Astolto and coworkers [44] aimed at evaluating the association between seasonality and BP onset through the number of hospitalizations recorded over the course of individual months. Although the date of BP onset and the date of hospitalization of BP patients could be somewhat different, data from D’Astolto and coworkers showed peaks for the exacerbation of BP in March and April (for women and men, respectively) and in June and July (for men and women, respectively). These findings were in line with the monthly distribution described by us for the pre-vaccination years (2018–2019–2020), which shows peaks in March and July. These data may possibly be explained by increased sun exposure and air temperatures but also by possible allergic reactions related to the spring peak [45,46,47]. Interestingly, the monthly distribution of BP onset during mass vaccinations was profoundly different from that observed during 2018–2020 (rs(10) = 0.05, 95% CI [−0.54, 0.61], *p* = 0.873), as well as from the one reported by D’Astolto et al. [44]. Specifically, in comparison with the three previous years, in 2021, we report a statistically significant increase in BP onset in April and May and a significant decrease in June and July, suggesting that, at least in some cases, there was an acceleration in the time to disease onset. Of note is that this “shifted” BP onset profile seems to be mainly due to VA patients rather than VNA patients. Interestingly, the correlation of the monthly distribution of BP onset in VA and VNA patients was positive and statistically significant (rs(10) = 0.59, 95% CI [0.00, 0.87], *p* = 0.049), suggesting that, at least in a portion of VNA patients, the monthly distribution of BP onset in 2021 was different from the one reported in previous years. A possible explanation could be the choice to discriminate between VA and VNA patients using an arbitrary, although reasonable, time of latency to BP onset (5 weeks), which is not capable of perfectly distinguishing between the two categories of patients: vaccine-associated and not-associated.

A valid speculation may be that vaccine-induced BP stems from vaccine-mediated stimulation of sub-clinical and pre-existent autoreactivity against BP180 and BP230. These autoantibodies have been detected in pruritic dermatoses of the elderly and in normal healthy individuals and could represent the autoimmune background overstimulated by vaccination [48,49,50,51,52]. In line with this speculation are our data on the relationship between anti-BP180 and BP230 autoAb titers and the time of latency from vaccination. In fact, higher titers correlated with shorter latency times, which may represent an indicator of association with vaccination (Appendix A).

It is important to note that the COVID-19 vaccine may act in a context of predisposition to BP: genetic susceptibility, comorbidities, and drugs may represent the fertile ground for disease onset, where a specific trigger (in this case, vaccination) is “the straw that breaks the camel’s back” [3]. In this study, no major difference was found in terms of comorbidities in the two groups (Appendix A); however, significant differences concerned the use of inhalers and diuretics (Table 3). The use of inhalers was higher in VA patients: it is possible to speculate that their use is linked to asthma and allergies [45,46,47], and this could also explain why the peak of BP onset is in the period of seasonal allergies, which could have contributed to the triggering of the disease. VNA patients had a higher intake of diuretics, especially loop diuretics and hydrochlorothiazide, both often reported as associated with BP onset [3,4,53]. The fact that diuretics were not involved in post-vaccination BP, while the association between BP and diuretics is widely reported, could suggest that in this subset of patients (VA), the triggering factors are of another nature, such as season-related. Moreover, a suitable environment induced by COVID-19 vaccines might make these subjects more prone to drug-induced BP, as has been reported in some cases of dipeptidyl peptidase IV inhibitors users [54,55,56].

## 5. Conclusions

In conclusion, to find an agreement between the relevant real-life experience of patients with a new-onset/exacerbation of BP following vaccination and other studies with conflicting results, COVID-19 vaccines may be considered as an accelerating factor that could likely induce autoimmunity, even without molecular mimicry, in genetically predisposed persons, by stimulating a pre-existent and subclinical autoreactivity against hemidesmosomal components. This phenomenon may slightly accelerate BP development without significantly modifying the overall disease incidence. Additional investigations on this controversial topic are needed, possibly within the framework of multicenter/multinational studies exploring, for instance, different settings, health systems, and cultural environments.

## Figures and Tables

**Figure 1 vaccines-12-01016-f001:**
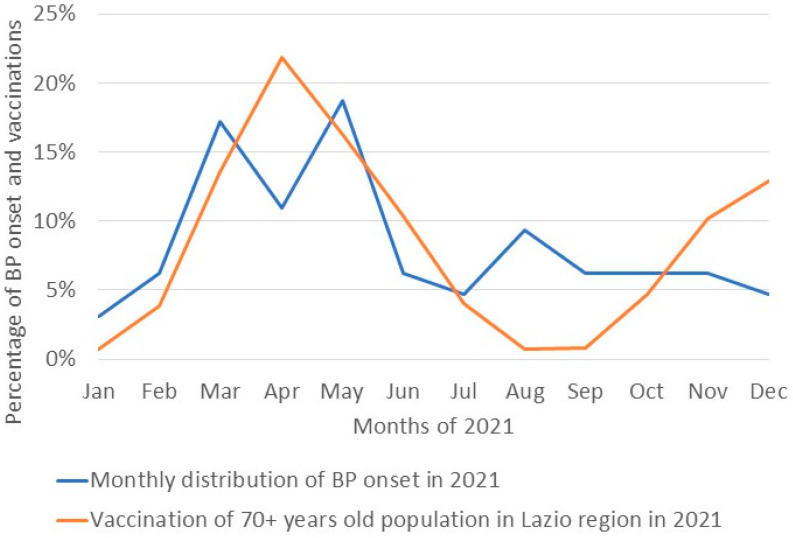
Monthly distribution of BP onset throughout the year paralleled distribution of COVID-19 vaccination. The blue line represents the monthly distribution of BP onset in our study population during year 2021. The orange line represents the distribution of COVID-19 vaccine administration in the general population over 70 years old in Lazio region, Italy. The peak of BP onset closely follows the peak of COVID-19 vaccinations in the general population.

**Figure 2 vaccines-12-01016-f002:**
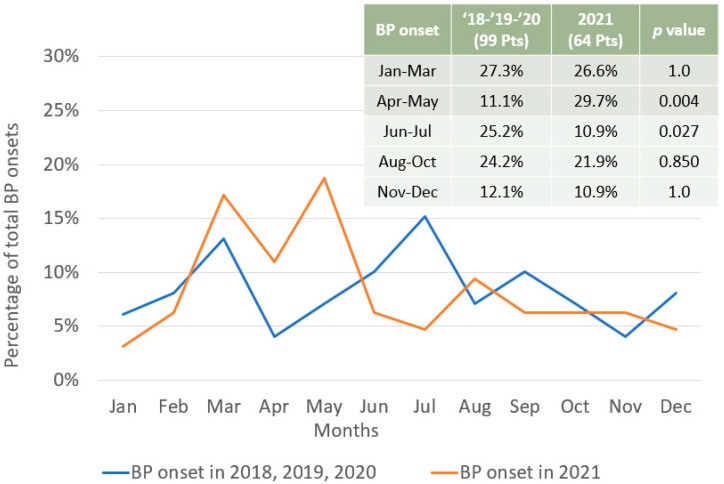
BP onset is differently distributed between pre-pandemic and pandemic conditions during COVID-19 mass vaccinations. Comparison of the monthly distribution of BP onset in 2018–2019–2020 (blue line, reported in the figure’s table as ’18–’19–’20) and in 2021 (orange line). *p*-values were calculated using Fisher’s exact (probability) test.

**Figure 3 vaccines-12-01016-f003:**
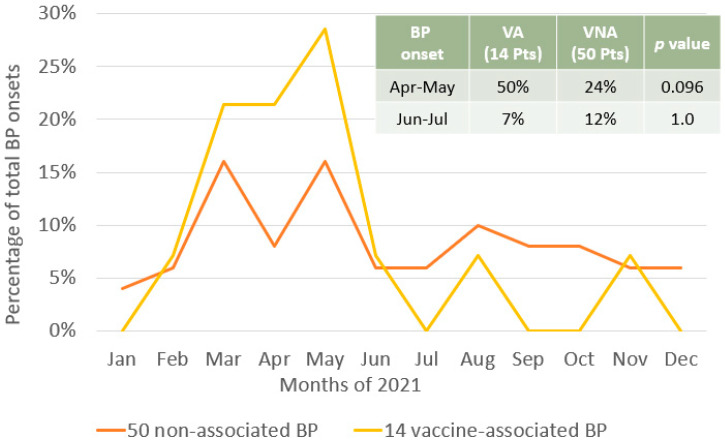
BP onset is differently distributed between vaccine-associated and vaccine-non-associated patients. Distribution of BP onset in 2021 in vaccine-associated (yellow line) and non-associated patients (orange line). *p*-values were calculated using the Fisher’s exact (probability) test.

**Table 1 vaccines-12-01016-t001:** Demographics and clinical and immunopathological features of 64 patients (14 VA and 50 VNA) with new-onset BP in 2021.

Patient Code	Sex, Age (y)	Vaccine	Histo- Pathology	DIF	IIF	BPDAI	ELISA BP180 (U/mL) Cut-Off ≥ 9	ELISA BP230 (U/mL) Cut-Off ≥ 9	ELISA 1080 (PIV) Cut-Off ≥ 14.9	ELISA 1331 (PIV) Cut-Off ≥ 4.5
14 VA patients	M/F ratio: 1.3 Mean age: 76.7 y	P = 10; Mod = 3; AZ + P = 1	ND = 3; Pos = 11	ND = 2; pos = 10; neg = 2	ND = 4; pos = 9; neg = 1	Mean: 28.2	pos = 13; neg = 1	pos = 7; neg = 7	pos = 4; neg = 10	pos = 3; neg = 11
BP1	M, 86	P/I^A^	ND	ND	pos	11.6	9.4	0.2	0.0	0.0
BP2	M, 72	P/I^A^	pos	pos	pos	10.0	259.9	0.0	15.4	0.0
BP3	M, 91	P/II^B^	ND	ND	pos	33.3	57.3	60.5	15.7	0.0
BP4	M, 73	P/II^B^	pos	pos	pos	53.0	1182.1	3.2	10.5	163.4
BP5	M, 80	P/II^B^	pos	pos	ND	3.6	59.2	2.0	101.0	0.0
BP6	M, 75	Mod/I^A^	pos	pos	pos	31.0	2.4	8.6	0.0	114.2
BP7	M, 89	Mod/I^A^	ND	pos	pos	17.6	38.9	63.0	0.0	0.0
BP8	F, 62	P/II^B^	pos	pos	neg	35.0	55.6	16.1	44.6	3.1
BP9	F, 79	P/I^A^	pos	pos	ND	6.5	30.3	1.6	0.0	0.0
BP10	F, 84	Mod/I^A^	pos	pos	ND	4.0	35.5	48.2	0.0	0.0
BP11	M, 74	P/II^B^	pos	neg	pos	40.0	43.4	15.3	0.0	104.2
BP12	F, 73	AZ (I), P (II)/I^A^	pos	neg	ND	0.0	9.1	6.4	0.0	0.0
BP13	F, 78	P/I^A^	pos	pos	pos	103.0	790.2	81.4	0.0	0.0
BP14	F, 58	P/II^B^	pos	pos	pos	46.0	922.6	14.9	0.0	0.0
50 VNA patients	M/F ratio: 1.3 Mean age: 80.0 y	*p* = 35; Mod = 5; AZ = 2; UK = 3; ND = 5	ND = 5; pos = 39; neg = 6	ND = 8; pos = 39; neg = 3	ND = 11; pos = 20; neg = 19	Mean: 32.9	pos = 38; neg = 12	pos = 22; neg = 28	ND = 1; pos = 19; neg = 30	ND = 1; pos = 18; neg = 31
BP15	M, 79	P (I-II), Mod (III)	pos	pos	pos	35.0	55.5	25.0	246.0	119.9
BP16	F, 81	P	pos	pos	neg	20.0	5.0	0.2	0.0	79.6
BP17	F, 91	Mod	ND	ND	pos	88.0	24.0	22.9	55.5	46.2
BP18	F, 83	P	pos	neg	pos	38.0	44.4	42.0	0.0	0.0
BP19	F, 87	P	pos	pos	ND	51.3	8.7	152.1	24.6	0.0
BP20	F, 72	P	pos	pos	ND	24.5	34.6	4.1	218.4	16.6
BP21	F, 74	ND	pos	pos	pos	32.7	69.2	2.8	195.3	67.2
BP22	M, 88	Mod	pos	pos	neg	45.0	1045.2	90.1	2.8	0.0
BP23	M, 82	P	neg	ND	pos	11.0	0.9	0.8	198.7	0.0
BP24	F, 68	P	pos	pos	neg	13.8	65.5	4.4	24.2	49.7
BP25	M, 90	P	pos	pos	ND	30.9	88.1	74.2	30.2	0.7
BP26	F, 82	P	pos	pos	neg	51.0	1367.3	6.1	0.0	29.2
BP27	F, 87	ND	pos	pos	neg	35.6	27.1	19.3	0.0	13.9
BP28	M, 90	P	pos	pos	neg	23.6	3.2	1.3	0.0	89.3
BP29	F, 76	P	pos	pos	pos	45.0	25.2	2.4	206.2	0.0
BP30	F, 72	P	pos	pos	pos	42.0	270.2	74.5	246.6	5.1
BP31	M, 82	P	pos	pos	pos	33.9	63.0	18.7	2.0	0.0
BP32	M, 74	P	pos	pos	neg	33.0	223.5	1.0	0.0	0.0
BP33	M, 89	P	pos	ND	pos	17.0	0.4	14.8	0.0	0.0
BP34	M, 83	ND	pos	neg	neg	21.0	93.6	26.4	136.5	6.6
BP35	M, 76	ND	pos	pos	neg	67.3	620.1	2.7	95.8	107.0
BP36	F, 72	AZ (I-II), P (III)	pos	pos	neg	50.3	305.5	0.7	0.0	0.0
BP37	M, 84	P	pos	pos	neg	19.6	58.6	2.0	0.0	0.0
BP38	M, 83	P	pos	pos	neg	27.0	95.8	62.2	0.0	4.6
BP39	M, 70	P	pos	pos	ND	UK	13.1	24.4	0.0	0.0
BP40	M, 82	P	pos	pos	neg	43.0	315.4	38.7	1.8	0.0
BP41	M, 82	P	pos	pos	pos	0.0	10.6	16.1	0.0	0.0
BP42	M, 80	P	neg	pos	pos	24.0	1.7	0.2	34.7	2.3
BP43	M, 61	P	pos	pos	ND	16.0	206.2	236.4	9.5	0.0
BP44	F, 75	P	pos	pos	pos	35.0	62.0	1.0	310.2	0.0
BP45	F, 85	P (I-II), Mod (III)	pos	pos	pos	28.6	41.4	50.9	97.9	0.0
BP46	M, 67	P	pos	pos	neg	75.0	2.9	1.2	0.0	59.3
BP47	M, 79	Mod	pos	pos	neg	29.0	0.8	13.5	0.0	0.0
BP48	M, 79	P	pos	ND	pos	65.0	9.6	2.3	326.4	7.7
BP49	M, 81	P	neg	pos	ND	28.0	1.0	22.9	0.0	0.0
BP50	F, 89	UK	pos	pos	pos	35.0	44.9	63.2	0.0	0.0
BP51	F, 87	UK	pos	pos	pos	29.6	33.0	0.3	0.0	0.0
BP52	M, 91	P	neg	pos	neg	125.0	186.7	0.6	68.8	47.7
BP53	M, 90	Mod	pos	pos	ND	27.0	143.7	0.7	231.9	70.7
BP54	F, 79	AZ (I-II), Mod (III)	neg	pos	neg	40.0	36.1	1.2	29.0	13.0
BP55	F, 69	P (I-II), Mod (III)	pos	pos	pos	42.0	32.8	1.4	0.0	0.0
BP56	M, 90	P	ND	ND	pos	24.0	2.2	2.9	0.0	0.0
BP57	M, 90	P	pos	pos	neg	30.0	89.0	1.6	0.0	0.0
BP58	F, 79	P	ND	ND	pos	0.0	108.2	0.2	ND	ND
BP59	M, 74	ND	neg	pos	ND	29.0	168.3	228.6	1.1	3.3
BP60	M, 81	P (I-II), Mod (III)	ND	pos	ND	0.0	0.9	2.0	0.0	0.0
BP61	F, 82	P	pos	neg	ND	0.0	44.4	54.2	3.7	0.0
BP62	F, 86	UK	pos	pos	neg	10.0	0.0	0.3	0.2	0.0
BP63	M, 47	P	pos	ND	ND	6.0	18.9	0.6	5.7	0.0
BP64	F, 81	Mod	ND	ND	pos	13.0	174.5	1.2	0.0	0.0

AZ: Astrazeneca; F: female; M: male; Mod: Moderna; ND: not done; neg: negative; P: Pfizer; pos: positive; UK: unknown; VA: vaccine-associated; VNA: vaccine-non-associated; y: years. I, II, and III indicate the first, second, and third vaccine dose, respectively. I^A^, the patient developed BP after the first dose; II^B^, the patient developed BP after the second dose.

**Table 2 vaccines-12-01016-t002:** Clinical and immunological features of vaccine-associated and vaccine-non-associated patients enrolled at diagnosis.

	8 VA Patients	34 VNA Patients	*p* Value
**BPDAI score (mean)**	41.1	37.2	0.669
**BP180 positivity (%)**	87.5	79.4	>0.999
**BP180 titer (median U/mL)**	259.9	88.1	0.335
**BP230 positivity (%)**	62.5	47.0	0.697
**BP230 titer (median U/mL)**	60.5	46.5	0.603
**E-1080 positivity (%)**	37.5	44.1	>0.999
**E-1080 titer (median PIV)**	15.7	95.8	** *0.039* **
**E-1331 positivity (%)**	25.0	41.1	0.688
**E-1331 titer (median PIV)**	138.8	46.9	** *0.033* **

VA: vaccine-associated; VNA: vaccine-non-associated. “BP180” only refers to the immunodominant region of BP180 (NC16A). *p*-values for BPDAI scores and autoantibody titers were calculated using the Mann–Whitney U test. *p*-values for reactivity to different epitopes were calculated using Fisher’s exact (probability) test. Statistically significant results are reported in bold and italic.

**Table 3 vaccines-12-01016-t003:** Percentage of vaccine-associated and vaccine-non-associated patients treated with different medications.

Drugs	50 VNA Patients	14 VA Patients	*p* Value
Antihipertensives (without diuretics)	74.0%	78.6%	>0.999
Diuretics (any)	50.0%	14.3%	** *0.030* **
Loop diuretics Hydrochlorothiazide Spironolactone Chlortalidone Indapamide Potassium canrenoate	30.0% 16.0% 4.0% 2.0% 2.0% 6.0%	7.1% 7.1% 0.0% 0.0% 0.0% 0.0%	0.159 0.670 >0.999 >0.999 >0.999 >0.999
Antihipertensives (any)	78.0%	85.7%	0.715
Cardiovascular medications (any)	70.0%	64.3%	0.749
Anticoagulant/antiplatelet agents (any) Aspirin Clopidogrel Warfarin	44.0% 26.0% 8.0% 12.0%	57.1% 28.6% 28.6% 0.0%	0.546 >0.999 0.062 0.206
CNS agents (any)	28.0%	35.7%	0.742
Anti-psychotics Anticonvulsants Anti-Parkinson/dopaminergic drugs Anti-dementia medications Benzodiazepins Anti-depressants	6.0% 8.0% 4.0% 2.0% 6.0% 6.0%	0.0% 28.6% 0.0% 0.0% 7.1% 7.1%	>0.999 0.062 >0.999 >0.999 >0.999 >0.999
Endocrine therapies (any)	52.0%	50.0%	>0.999
DPP4-i	28.0%	14.3%	0.487
Antitumorals (anti-PD1)	4.0%	14.3%	0.206
Gastrointestinal medications	44.0%	28.6%	0.367
Inhalers	0.0%	14.3%	** *0.0* ** ** *45* **
Prostatitis and urinary tract infection drugs	14.0%	7.1%	0.673
Ophtalmic drugs	0.0%	7.1%	0.219
Analgesics/NSAIDs	8.0%	7.1%	>0.999
Allopurinol	20.0%	0.0%	0.101
CRI medications	4.0%	0.0%	>0.999

CNS: Central Nervous System; CRI: chronic renal insufficiency; DPP4-i: dipeptidyl peptidase 4 inhibitors; NSAIDs: non-steroidal anti-inflammatory drugs; VA: vaccine-associated; VNA: vaccine-non-associated. *p*-values were calculated using Fisher’s exact (probability) test. Statistically significant results are reported in bold and italic.

## Data Availability

The raw data supporting the conclusions of this article will be made available by the authors on request.

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
