# Peer review of "COVID-19 Vaccine: A Potential Risk Factor for Accelerating the Onset of Bullous Pemphigoid"

_vaccines, 2024, doi:10.3390/vaccines12091016_

Round 1

Reviewer 1 Report

Comments and Suggestions for Authors

Many minor revisions are reccomended. Although the manuscript is of high medical importnance certain issues are not heavili stated. 

For instance, the authors say in introduction: Nonetheless, the many real-world clinical data pointing to the necessity to further investigate this issue should not be ignored

They should consider to change this phrase.

Also, what does this mean by being ignored? Previously the authors have mentioned sufficient studies involving 92 BP cases Ref 10.

Why do authors are underestimating this previous study?

This kind of thinking should go along the whole of manuscript. Authors should reconsider the magnitude of mRNA vaccination causing autoimmunity. What about the T regulatory response these  patients have?

Specific revisions

o   A special paragraph / small section dedicated on autoimmunity caused by the mRNA vaccinations should be added. A short table list and correlations of these autoimmune phenomena with BP autoimmune disorder should be evaluated.

o   Why do the elderly vaccinated individuals show a higher trend of developing BP as shown in figure 1?

o   Authors need to address the immune-senescence phenomena caused by the mRNA COVID-19 vaccination. There are sufficient published data. Also, for other autoimmune skin disorders caused by the mRNAs, like psoriasis.

For other drug associations with BP onset.

Authors should consider why the vaccine associated drugs increase the prevalence of BP occurrence. I do believe that a thorough comment should be addressed.

For instance, why the vaccine associated patients take more anticoagulants and other drugs.

I suggest authors should consider to change this pattern of paper state the importnace of developing autoimmunity after the mRNA vaccinations and then reconsider to proceed to other minor revisions if their response is not thorough and adequate.   

Author Response

Reviewer: Many minor revisions are recommended. Although the manuscript is of high medical importance certain issues are not heavily stated.

Reply: We would like to thank the reviewer for his/her appreciation.

Reviewer: For instance, the authors say in introduction: “Nonetheless, the many real-world clinical data pointing to the necessity to further investigate this issue should not be ignored”. They should change this phrase. Also, what does this mean by being ignored? Previously the authors have mentioned sufficient studies involving 92 BP cases Ref 10. Why are authors underestimating this previous study?

Reply: We thank the reviewer for the insightful consideration, that led us to change this sentence in order to highlight our underlying opinion (lines 80-82). We do believe real-life data are crucial in assessing the association between vaccinations and bullous pemphigoid onset, as we stated in some of our previous studies (references cited in the current paper as 8, 33, 43, 56).

Reviewer: This kind of thinking should go along the whole of manuscript. Authors should reconsider the magnitude of mRNA vaccination causing autoimmunity.

Reply: We agree with reviewer’s comment. We are convinced that vaccination plays a role in the induction of autoimmunity and the present study confirms this point. We rephrased several sentences throughout the paper to underline the importance of the clinical data of patients experiencing disease onset following vaccination in the context of our work (lines 302-303, 375, 401, 403, 404, 406). In this context, see also Supplementary Fig 1 and lines 57-68, 217-220, 225-227 and 379-382.

Reviewer: What about the T regulatory response these patients have?

Reply: Unfortunately, we did not collect the biological material necessary to assess the T regulatory response of these patients, which, therefore, has not been investigated.

Reviewer: A special paragraph / small section dedicated on autoimmunity caused by the mRNA vaccinations should be added. A short table list and correlations of these autoimmune phenomena with BP autoimmune disorder should be evaluated.

Reply: following the reviewer’s advice, we provided better background information about the link between COVID-19 vaccinations and autoimmunity and the possible mechanisms leading to autoimmunity and BP (lines 57-68). We also added a short table list of autoimmune diseases that have been documented following the administration of COVID-19 vaccines (Supplementary Table 1).

Reviewer: Why do the elderly vaccinated individuals show a higher trend of developing BP as shown in figure 1? 

Reply:  We would like to thank the reviewer for this observation, that shows us that Figure 1 legend is not clear enough. In fact, Figure 1 represents the comparison of the monthly distribution of BP onset in our study population and COVID-19 vaccination during 2021 and does not illustrate a higher trend of developing BP in elderly vaccinated individuals. BP typically affects the elderly, and about 90% of our study population is over 70 years old, hence the reason we only reported vaccinations in the general population for this age class. The crucial point underlined in this figure is the maximum of percentage of BP onset that paralleled the peak of vaccinated individuals. We modified the legend of Figure 1 in order to make it clearer.

Reviewer: Authors need to address the immune-senescence phenomena caused by the mRNA COVID-19 vaccination. There are sufficient published data. Also, for other autoimmune skin disorders caused by the mRNAs, like psoriasis.

Reply:  In line with reviewer’s suggestion, we discuss the immune-senescence phenomenon and autoimmune skin disorders as consequence of COVID-19 vaccination in lines 57-68.

Reviewer: For other drug associations with BP onset. Authors should consider why the vaccine associated drugs increase the prevalence of BP occurrence. I do believe that a thorough comment should be addressed. For instance, why the vaccine associated patients take more anticoagulants and other drugs.

Reply: The only statistically significant data concerned the higher and lower use of inhalers and diuretics, respectively, in vaccine associated vs vaccine non-associated patients. Additional statistical analyses were performed (see Supplementary Figure 2 and lines 257-274). Unfortunately, the small size of the study population hinders the production and interpretation of conclusive data on drugs, and we believe studies with larger cohorts are necessary to discuss the impact of drugs on vaccine-associated BP.

Reviewer: I suggest authors should consider to change this pattern of paper to state the importance of developing autoimmunity after the mRNA vaccinations and then reconsider to proceed to other minor revisions if their response is not thorough and adequate.  

Reply: in line with the requests from Reviewer 1, we modified the manuscript stating the importance of developing autoimmunity after the mRNA vaccinations. Moreover, we proceeded with all others requested revisions.

Reviewer 2 Report

Comments and Suggestions for Authors

Pira's paper entitled "COVID-19 Vaccine: A Potential Risk Factor for Accelerating the Onset of Bullous Pemphigoid" has valuable information, but lacks rigorous statistical analysis.

·         From the data in Figures 1 through 3, calculate the Spearman or Pearson correlation coefficients (whichever is appropriate, depending on the distribution) and the interval coefficient of r

·         Compute a multivariate correlation matrix from the data in Table 3

·         Using the data in Table 2 and the data in the supplemental file, run a binary logistic regression.

·         L83.- It is necessary to add antibody information to the criterion.

Besides

·         Authors need to provide more information on antibodies for BP180, BP230, 1080, 1331, L107.

·         Of the patients who developed BP, what was the antibody titer at baseline and throughout the course of the disease?

·         L141.-Provide information about BPDAI

·         Limitations of the work must be stated by the authors.

Author Response

Reviewer: Pira's paper entitled "COVID-19 Vaccine: A Potential Risk Factor for Accelerating the Onset of Bullous Pemphigoid" has valuable information, but lacks rigorous statistical analysis.

Reply: we thank the reviewer for the appreciation and for their advice, which led us to improve statistical analyses. 

Reviewer: From the data in Figures 1 through 3, calculate the Spearman or Pearson correlation coefficients (whichever is appropriate, depending on the distribution) and the interval coefficient of r

Reply: We appreciate this suggestion. We computed Spearman’s correlation coefficients with data from Figures 1-3 and reported them in the results section (lines 217-220, 225-227, 229-231). We also addressed these results in the discussion (lines 348-349, 362, 367-374). We also included the test in the pertaining section of Materials and Methods (lines 126-127). 

Reviewer: Compute a multivariate correlation matrix from the data in Table 3

Reply: To uncover patterns and relationships within the drug dataset, data reduction strategy, including Principal Component Analysis (PCA) and heatmap have been used. The data are displayed in Supplementary Figure 2, and reported at lines 257-274.  We also included the test in the pertaining section of Materials and Methods (lines 130-132). For this analysis, we used the expertise of Dr. Scaglione, who has been added to the author list. We were unable to perform further statistical analyses due to the small sample size and, additional data are required to fulfil this task.

Reviewer: Using the data in Table 2 and the data in the supplemental file, run a binary logistic regression.

Reply: Thanks to the reviewer’s suggestion, we ran a binary logistic regression (Supplementary Table 4) using the data in Table 2 and in the supplemental file with the list of comorbidities (which is now Supplementary Table 3) and reported the data in the material and methods and results sections (lines 127-130, 276-282). As well as the previous one, also this analysis was made by Dr. Scaglione.

Reviewer: L83.- It is necessary to add antibody information to the criterion.

Reply: in line with the request, we changed the criterion (lines 92-93, 110).

 Reviewer:  Authors need to provide more information on antibodies for BP180, BP230, 1080, 1331, L107.

Reply: Thanks to this comment we had the opportunity to better delineate the autoantibody profiles in the different groups of patients by adding information in the results (lines 175-183). Additionally, to avoid any possible misinterpretations, we modified Table 2 and rephrased a brief sentence in the materials and methods section (lines 113-118).   

Reviewer:  Of the patients who developed BP, what was the antibody titer at baseline and throughout the course of the disease?

Reply: We thank the reviewer for this question. We refined the information about autoantibodies by changing Table 1 to report the titers, instead of only the positivity or negativity, for each one of the 64 patients who developed BP. The antibody titers were measured on serum samples collected the day the patients were enrolled in the study, and were not measured afterwards as follow-up visits were not planned for this study, therefore we do not have any information about their trend during the course of the disease. This has been stated as a limitation of this study (lines 306-309).

Reviewer: L141.-Provide information about BPDAI

Reply: we appreciated this suggestion. Information about BPDAI was included in the text (lines 191-199) with an appropriate reference. Moreover, patients’ data were enriched by adding the BPDAI scores at enrollment in Table 1 and their mean values in Table 2 and Supplementary Table 2.  

Reviewer: Limitations of the work must be stated by the authors.

Reply: we agree with the reviewer and stated the limitations of our study in the discussion (lines 306-309).

Reviewer 3 Report

Comments and Suggestions for Authors

I have reviewed the paper by Pira et al.

Authors need to provide citations for statements like “ Bullous pemphigoid (BP) is the most common autoimmune blistering disease (AIBD)”.

The article is important, as authors explain, since prediction of autoimmune diseases following COVID-19 vaccination has been predicted.

Authors need to provide a rationale about the 5 weeks to dichotomize “vaccine associated” vs. “non-vaccine associated”. In my opinion, the risk associated with endosomal TLR signaling may take even years, as in the case of SLE. It would make much sense to compare “patients not vaccinated” as VNA. Although in their cohort these are just n=5, an analysis should be made.

Also, an analysis of mRNA vaccine compared to virus-based (Astra-Zeneca) needs to be done, as I presume the mRNA vaccines bring potentially more risk. Authors state in the Discussion that “The differences in vaccine type administration in VA and VNA patients were not significant, seemingly indicating that vaccine types are not associated with different risks of developing BP.”, but I have not seen any analysis to support this.

I like the Table 2, but it should be re-analyzed with VNA Non-vaccinated only. It would be great to accompany it with a correlation plot with all titers and showing to groups (VA vs VNA).

What I don’t see is a plot of “Time since vaccination” of patients with bullous pemphigoid (and their titers). This would help backup the indirect (and very suggestive data) from Figures 1-3.

Data from Table 3 can be clarified by running a multivariate analysis.  

Author Response

Reviewer: I have reviewed the paper by Pira et al. Authors need to provide citations for statements like “Bullous pemphigoid (BP) is the most common autoimmune blistering disease (AIBD)”.

Reply: we have inserted an appropriate reference at line 35 (ref 1) and line 51 (ref 5,6).  

Reviewer: The article is important, as authors explain, since prediction of autoimmune diseases following COVID-19 vaccination has been predicted.

Reply: we thank the reviewer for the appreciation of our work.

Reviewer: Authors need to provide a rationale about the 5 weeks to dichotomize “vaccine associated” vs. “non-vaccine associated”. In my opinion, the risk associated with endosomal TLR signaling may take even years, as in the case of SLE.

Reply: we agree with the reviewer that a causative relationship between vaccine and BP onset could be present also after years from the date of vaccination. Notwithstanding, in the present study we consider short term effects of vaccination, because for long term effects a retrospective study after 5-10 years from the time of mass vaccinations on vaccinated and not vaccinated patients should be conducted. However, for a probable association between vaccine and BP onset a reasonable time-related continuity after vaccine application should be considered. We consider a latency time of 5 weeks a time sufficient both for stimulation of pre-existent autoimmunity and for the induction of a novel autoimmune response triggered by the vaccine. Moreover, as previously shown, a one-month latency period from the time of vaccination may be appropriate for an autoimmune response to hemidesmosomal antigens to be developed (see ref 38). This has been clarified in the Materials and Methods section and an appropriate reference has been added (lines 97-102, ref 38).

Reviewer. It would make much sense to compare “patients not vaccinated” as VNA. Although in their cohort these are just n=5, an analysis should be made.

Reply: We agree with the reviewer’s observation that a comparison between BP vaccinated and not vaccinated patients could be extremely interesting. However, considering that BP are mainly over 70 years old and mass vaccination in Italy involved the majority of over 70 individuals, not vaccinated BP were rare in our cohort (only 5 patients). Although the number of not-vaccinated BP patients is limited, we have now performed some analyses on immunological profiles and drugs use that involve this subgroup of patients (see lines 171, 184-188, 197-199, 328-329, 338-341 and Supplementary Table 2 and Supplementary Figure 2) as requested.

Reviewer: Also, an analysis of mRNA vaccine compared to virus-based (Astra-Zeneca) needs to be done, as I presume the mRNA vaccines bring potentially more risk. Authors state in the Discussion that “The differences in vaccine type administration in VA and VNA patients were not significant, seemingly indicating that vaccine types are not associated with different risks of developing BP.”, but I have not seen any analysis to support this.

Reply: we agree with the reviewer that the section regarding the different types of vaccinations needs to be clarified. We rephrased the results about vaccination and added all the information about the vaccines administered in each group (lines 144-159, 325-326). Moreover, the reviewer’s comment led us to carefully re-esaminate Table 1, giving us the opportunity to discover some typos and minor mistakes that have been corrected both in the Table and in the text (lines 142-144), although they did not impact the meaning of the work.

Reviewer: I like the Table 2, but it should be re-analyzed with VNA Non-vaccinated only. It would be great to accompany it with a correlation plot with all titers and showing to groups (VA vs VNA).

Reply: We thank the reviewer for the suggestion. We improved the analysis of clinical and immunological profiles by adding a comparison between VA patients, all vaccinated patients and non-vaccinated patients only (lines 171, 184-188) and reported the data in a dedicated Table (Supplementary Table 2). As the data we obtained are not statistically significant, we decided against adding a correlation plot to the study.   

Reviewer: What I don’t see is a plot of “Time since vaccination” of patients with bullous pemphigoid (and their titers). This would help backup the indirect (and very suggestive data) from Figures 1-3.

Reply: We agree with the reviewer that this plot would be very interesting. We added a plot of time since vaccination and autoAb titers (Supplementary Figure 1) and reported the correlation in the results section (lines 200-206) and in the discussion (lines 379-382). However, it should be considered that to accurately assess the effect of vaccinations, autoAbs titers should only be the ones of patients enrolled at diagnosis (i.e. patients who were not treated for BP at enrollment), and we have a limited number of these patients. Of the 34 VNA patients at disease diagnosis, 5 non-vaccinated individuals and 10 patients who received at least one vaccine dose after BP onset had to be excluded from this analysis, which was conducted on 27 vaccinated patients with disease onset following vaccination.

Reviewer: Data from Table 3 can be clarified by running a multivariate analysis.  

Reply: To uncover patterns and relationships within the drug dataset, data reduction strategy, including Principal Component Analysis and heatmap have been used. The data are displayed in Supplementary Figure 2, and reported at lines 257-274.  We also included the test in the pertaining section of Materials and Methods (lines 130-132). For this analysis, we used the expertise of Dr. Scaglione, who has been added to the author list. We were unable to perform further statistical analyses due to the small sample size and, additional data are required to fulfil this task.

Round 2

Reviewer 2 Report

Comments and Suggestions for Authors

Indicate the statistical test at the bottom of the tables.

The statistical analysis of their work has been improved by the authors.

Author Response

Reviewer: Indicate the statistical test at the bottom of the tables.

Reply: we added the tests used to calculate p-values in the manuscript at the bottom of Table 2 and Table 3, as well as in Figure 2 and Figure 3. We also added the information at the bottom of Supplementary Tables 2, 3 and 4. This edits also gave us the opportunity to discover some missing abbreviations, which have now been added, in Table 3 and Supplementary Table 4.

Reviewer: The statistical analysis of their work has been improved by the authors.

Reply: we thank the reviewer for the appreciation of our work.

Reviewer 3 Report

Comments and Suggestions for Authors

My comments have been addressed satisfactorily.

Author Response

Reviewer: My comments have been addressed satisfactorily.

Reply: we thank the reviewer for the appreciation of our work.